# Effect of Relocation, Social Housing Changes, and Diarrhea Status on Microbiome Composition of Juvenile Cynomolgus Macaques (*Macaca fascicularis*)

**DOI:** 10.3390/microorganisms13010098

**Published:** 2025-01-07

**Authors:** Keely McGrew, Nicole Monts de Oca, Therese A. Kosten

**Affiliations:** 1Charles River Laboratories, Inc., Houston, TX 77047, USA; keely.mcgrew@crl.com (K.M.); nicole.montsdeoca@crl.com (N.M.d.O.); 2Department of Psychology, University of Houston, Houston, TX 77004, USA

**Keywords:** dysbiosis, gut–brain axis, microbiome, primates, relocation stress

## Abstract

Social housing changes are likely stressful and can be associated with diarrhea, the most common health problem noted in captive macaque populations. Diarrhea may reflect a negative shift in the gut flora (“gut dysbiosis”). This study reported on changes in the gut microbiome composition of juvenile primates (*Macaca fascicularis*) that experienced a change in social housing and exhibited diarrhea. A matched-case–control design was utilized to compare fecal samples from gut-unhealthy animals to healthy counterparts (*n* = 61). Baseline samples from recently imported animals were collected during routine sedation events. When an animal experienced a housing change, the entire cohort was monitored for diarrhea. Post-relocation samples were collected from animals that exhibited diarrhea and from their matched controls. Samples were assessed via 16S rRNA next-generation sequencing for a microbiome analysis and by ELISA for cortisol levels. Fecal cortisol levels did not differ between groups or across time points. Alpha diversity increased after relocation and differed by sex with males demonstrating a greater change in alpha diversity (*p* < 0.01). Although exhibiting diarrhea did not affect alpha diversity levels, it was associated with increased beta diversity (*p* < 0.05). Understanding how the microbiome may be affected by relocation will help guide prevention strategies such as the use of specific probiotics to reduce the incidence of diarrhea.

## 1. Introduction

Diarrhea is considered one of the most common and most challenging health problems in macaque colonies around the world, reported to affect as much as 15–39% of the population at various facilities [1,2]. Studies evaluating the pathogenesis, causes, and success of various treatment modalities have been conducted for several years [3,4,5]. Yet, chronic diarrhea that is unresponsive to treatment is still one of the leading causes of mortality in macaques [6,7,8]. Although many bacterial and parasitic pathogens have been identified to cause diarrhea in macaques, a significant number of cases are considered idiopathic and likely multifactorial [9,10,11]. Stressful situations such as relocation increase risk of diarrhea in macaques [9]. Evaluating the effects that factors such as stressful situations have on the development of diarrhea in macaques continues to be an area of focus for the research community.

Changes within group hierarchies can be stressful to macaques because they are social animals with a linear hierarchy and a range of temperaments [9,12]. These social changes can cause alterations in the stress pathways that can last for several days [13]. In addition, social disruption can lead to alterations in the hosts’ microbial communities and impacts to their immune system [14,15,16]. Mental and emotional states, such as the perceived stress of relocation and associated social hierarchy changes, have bi-directional effects between the hypothalamus–pituitary–adrenocortical (HPA) axis and the gut that contribute to a negative shift in the gut microbiome, referred to as “gut dysbiosis” [17,18,19]. These effects on the HPA axis result in the release of glucocorticoids such as cortisol [17,18,20].

When in a state of dysbiosis, an organism may experience “leakiness” of the gut, the malabsorption of nutrients [21], metabolic changes [22,23], and diarrhea. The relationship between stress-related “gut dysbiosis” and diarrhea has been documented in many species of animals including dogs [24], cats [25], pigs [26], and cattle [27], as well as humans [28]. Although studies in nonhuman primates (NHPs) have linked changes in GI health to differences in gut microbiome communities [22,29], to the best of the authors’ knowledge, no studies have evaluated the association between a relocation event and associated social disruption, diarrhea, and the gut microbiome composition of macaques.

In the Mauritian colony of cynomolgus macaques (*Macaca fascicularis*) housed at the Charles River Import/Export facility in Houston, diarrhea rates increase when the animals are moved from the quarantine building that holds animals in pairs in standard quad caging to a separate building composed of group housing units of 8–10 individuals. We hypothesize that the social housing transitions cause individual animals to experience a “gut dysbiosis” that contributes to the incidence of diarrhea. Thus, we tested whether animals that exhibit diarrhea after relocation would show enhanced cortisol levels and alterations in the gut microbiome. The aim of this study was to characterize the connection between a relocation event, a physiological response to stress, and changes to the fecal microbiome composition. The information gained from this study would be useful in determining strategies for managing gut health in primates experiencing relocation events.

## 2. Materials and Methods

### 2.1. Subjects and Housing

We employed a matched-case–control design to compare gut-unhealthy animals (i.e., those that exhibited diarrhea after relocation) to those from the same cohort that appeared to be gut-healthy (i.e., did not exhibit diarrhea after relocation) by assessing fecal samples collected prior to and after housing relocation. The sample size was powered sufficiently to probe for potential sex differences. The connection between a social housing change, a physiological stress response, and an unhealthy microbiome composition (“dysbiosis” and/or diarrhea) has not been specifically documented in the literature. Therefore, characterizing these connections with the intention of utilizing this information in identifying preventative or therapeutic interventions would improve animal welfare and could positively impact overall animal husbandry practices.

Cynomolgus macaques of both sexes imported from the island of Mauritius to Charles River’s import–export site during May and June 2020 were included in this study. The animals included in the microbiome analysis portion of the study (*n* = 61) were all from the same farm and arrived in the facility close to the same time. All animals were juveniles, in the age range of 1.89–3.81 years (mean: 2.97, sd: 0.79) at the beginning of the study. The subjects ranged in weight from 2.1 to 4.4 kg (mean: 3.26 kg, sd: 0.67) at the beginning of the study.

All the animals were housed indoors. The animals were initially housed in a CDC-approved quarantine building in quad-rack cages, affording 4.69 sq ft of floor space per animal. Dividers located between the two cages in both upper and lower portions of the quad-rack caging were removed, allowing continuous pair housing in which two animals of the same sex shared two cage spaces. The animals were in quarantine for approximately 35 days. Following quarantine, most animals were relocated to another building, where they were housed in groups. Every effort was made to put animals in groups with their social partner from the quarantine period, and with other animals from the same quarantine room and that originated from the same farm. The group housing area is composed of 120 (6 × 8 × 6.5 ft) stainless steel social housing units. Each unit can house up to 11 monkeys that each weigh less than 10 kg per the USDA and NHP *Guide* housing recommendations [30]. Typically, animals are housed at a density of 8 individuals per group. Each unit includes structures such as perches at varying levels, swings, toys, or other devices to promote exercise and provide visual barriers, as outlined by our institute’s husbandry and care program.

Housing environments were maintained at 69 to 75 °F (22.6 to 23.9 °C), with relative humidity of 30% to 70%, and on a 12:12 h light/dark cycle. Chlorinated and filtered municipal water was provided ad libitum through an automated watering system. All animals were examined by a veterinarian at import, dosed with ivermectin and praziquantel (Droncit), and tested for tuberculosis. All the animals were fed a commercial monkey chow (Lab Fiber-Plus Monkey Diet, PMI International, Brentwood, MO, USA) throughout their entire time at the facility and given supplemental treats and fresh produce on a regular rotating basis. The Lab Fiber-Plus Monkey Diet includes at least 20% crude protein, 5% crude fat, and between 9 and 10% crude fiber, predominantly provided through soybean hulls. Reports of diarrhea upon arrival and throughout the quarantine period are very rare.

### 2.2. Ethics Statement

All procedures were approved by the Animal Care and Use Committee at Charles River’s Houston location under IACUC (Protocol # P01292020-H; 01/29/2020). The facility complies with the Animal Welfare Regulations (AWRs) and is AAALAC-accredited.

### 2.3. Study Design

Based on our prior observations of a 10% incidence rate of diarrhea observed after housing relocation in imported macaques, we chose a matched-case–control design, a preferred method for studying conditions with low incidence in the population [31]. In the absence of similar studies, effect size was unknown but a sample size of 30 matched cases (diarrhea and control) with two time points (before and after housing relocation) was estimated to be effective at determining a statistically significant effect at either a large or medium effect size. We attempted to gather samples from equal numbers of males and females.

When an animal was noted to show diarrhea after relocation, fecal samples from these gut-unhealthy animals along with their unaffected control from the same shipping cohort were collected. Fecal samples from all animals were collected at baseline and stored to be analyzed if the animal was selected to be studied. The experiment ended 8 weeks after the animals moved to the group housing unit.

### 2.4. Specimen Collection

Fecal samples of all animals of Mauritius-origin imports (112–120 animals) were collected approximately 6 days after arrival into the facility. Samples were collected by inserting a fecal loop into the rectum of the animal, and then relocating the feces into a cryotube using a cotton-tipped applicator. Samples were collected between 7 and 11 am and were stored at −80 °C until the samples were sent to a lab for the analysis. Samples from all incoming animals were collected and stored, as it was not known at the time of arrival which animals developed diarrhea. A second set of fecal samples, for use in a cortisol ELISA assay, was collected approximately four weeks after arrival during relocation to group housing. The animals had been living in continuous pair housing in this same room for four weeks prior to cortisol specimen collection. This time point was chosen for baseline cortisol assessment because by 30 days after the initial transportation event from Mauritius to Houston, the physiological impact of travel should have worn off with the cortisol levels returning to normal [32]. Fecal glucocorticoid (specifically cortisol) levels can represent the magnitude of a stressor in the past several hours to several days [33,34].

After each import cohort was moved to group housing, the cohort was monitored for diarrhea during weeks 2–8 post-relocation. Animals in the same housing unit were matched by age, weight, and sex, and were exposed to the same food, environment, and care staff. Every effort was made to minimize movements in and around those pens to minimize social disruption. Monitoring diarrhea consisted of reading daily health reports filled out by trained animal care technicians. If diarrhea was observed inside the social housing unit, the technicians recorded a fecal score (4 being loose stool, and 5 and 6 indicating progressively more liquid stool). Following a report of a fecal score of 5 or 6 in a group, the individual within the group with diarrhea was identified and a fecal sample was collected. A second fecal sample was collected from a healthy animal from the same social housing unit to serve as the control sample. The feces were placed in two separate 2 mL cryotubes (one for the microbiome analysis and one for the second cortisol time point), labeled, and placed in a −80 °C freezer. Upon the completion of the collection of all desired samples, the samples stored in the freezer were removed, and sorted, and those final samples plus their corresponding frozen baseline samples were processed for 16S sequencing or a cortisol assay.

### 2.5. Microbiome Sequencing

16S ribosomal RNA gene sequencing was conducted by the Research Animal Diagnostics Services group of Charles River Laboratories (Wilmington, MA, USA). Samples were processed following the Illumina-recommended 16S rDNA sequencing protocol. In brief, DNA was isolated from submitted samples using the MoBio/Qiagen Powerlyzer Powersoil kit per the manufacturer’s protocol (Thermo-Fisher, Waltham MA, USA). The workflow was based on Illumina’s 16S rRNA metagenomics workflow, including the cited V3-V4 primers with adapter sequences, with minor optimizations. Recovery yield and DNA quality were determined by a fluorometric analysis (Quibit). DNA concentration was adjusted to specifications (diluted to 5 ng/uL per reaction, with 5 uL used in the reaction for a total of 25 ng used in the amplicon reaction) and amplified using broadly reactive primers spanning the 16S rRNA gene V3 and V4 regions. Resulting amplified PCR products were analyzed for quantity and correct product size (Agilent Bioanalyzer 2100; Santa Clara, CA, USA), then purified and amplified with primers containing unique molecular barcodes (Illumina, San Diego CA, USA). Amplicon quality and quantity were further analyzed by SYBR green qPCR (KAPA). All samples were pooled and adjusted to a normalized concentration. The DNA library pool was denatured with sodium hydroxide, normalized to optimal loading concentration, and combined with PhiX control (Illumina). Libraries were sequenced on the Illumina MiSeq platform using the 2 × 300 bp paired-end protocol. After sequencing, the raw FASTQ data were uploaded into the One Codex bioinformatics portal, which provides 16S rRNA classification using their curated Targeted Loci Database (as the V3–V4 region of the 16S rRNA gene is a target loci). Per One Codex, “The Targeted Loci Database on the One Codex platform is specifically designed for marker gene analysis, with a large, curated database that includes 16S, ITS, and more”. Analyzing 16S data against this database provides highly accurate microbial identification; community diversity measures that are robust to sequencing depth; a completely reproducible analysis using a stable, versioned reference database; and large-scale cross-comparison across samples in the One Codex platform [35]. Species-level assignment was carried out using this Targeted Loci Database on the One Codex platform.

Following the taxonomic classification of microbiome samples, the Agile Toolkit for Incisive Microbial Analyses version 2 (ATIMA), an in-house R-based platform created by the Baylor College of Medicine Alkek Center for Metagenomics and Microbiome Research, was used for the data analysis.

To evaluate alpha diversity, a within-sample measure of ecological diversity, the Shannon Index and observed Operational Taxonomic Unit (OTU) metrics were used. Observed OTUs are a measure of the number of unique OTUs identified in a given sample, whereas the Shannon Index is an estimator of the richness and evenness of taxa in a sample, quantifying uncertainty in taxonomy identity [36]. Both measures were calculated using OTU-level data.

By comparison, beta diversity measures the relative dissimilarity between two communities or samples [37]. Beta diversity was visualized by a principal coordinates analysis (PCoA), and overall compositional differences between groups were assessed using the PERMANOVA statistic. To determine whether certain groups of animals experienced greater changes to their microbiome over time, we used each animal’s pairwise beta diversity score between the second sample (time of diarrhea outbreaks) and baseline and evaluated differences between groups using the Mann–Whitney U test in R version 4.0.3, an open-source statistical analysis platform [38].

### 2.6. Cortisol Measures

Cortisol was measured from fecal samples utilizing the Cayman Chemical Cortisol ELISA kit #500360 (Ann Arbor, MI, USA). This is a competitive assay that has a range from 6.6 to 4000 pg/mL and a sensitivity of approximately 35 pg/mL. The assay is based on a competition between cortisol and cortisol-acetylcholinesterase (AchE) for a limited number of cortisol-specific mouse monoclonal antibody binding sites and has been utilized in cortisol assays for NHPs [39]. For the extraction of cortisol, wet fecal samples were weighed to 0.1 g. The feces were added to 1 mL of 100% methanol (MeOH) in a plastic centrifuge tube and homogenized using a metal stirring rod. Each sample was vortexed for 30 s and then centrifuged for 10 min at 2500 rpm. The supernatant was removed and put into an Eppendorf tube using a micropipette. The sample was diluted (in MeOH) 1:20 using the Cayman EIA buffer (20 µL + 380 µL buffer) and then a standard protocol completed the assay (Cayman Chemical, Ann Arbor MI, USA). Microplates were placed on a Biotek absorption reader and analyzed to assess the amount of light absorption per sample using Gen5 version 2.2 software. Readings were recorded at 405, 410, and 420 nm. The blank values were too high at 405 and 410 nm and the most reliable readings were determined to be the ones read at 420 nm. Each sample was processed in duplicates and final concentrations for the duplicates were averaged for baseline and post-stress measurements for each animal.

### 2.7. Data Analysis

Most data were analyzed using ANOVA in R. Comparisons of taxa abundance levels between groups (Healthy vs. Unhealthy) in the matched sample set were analyzed with repeated measures ANOVA using the lm function in R with abundance as the dependent variable (DV). The Health Status group and Sex were the independent variables (Iv) with repeated measures on Time (baseline vs. relocation). Bonferroni’s correction was applied for multiple comparisons (adjusted *p* = 0.006). Cortisol levels were analyzed similarly.

ATIMA is a stand-alone tool for analyzing and visualizing microbiome data sets that combines publicly available packages [40,41] with purpose-written code to identify trends in taxa abundance, alpha diversity, and beta diversity as they relate to sample metadata. Categorical variables were evaluated using the non-parametric Mann–Whitney U [42] and Kruskal–Wallis tests [43] for variables with two groups or three or more groups, respectively. Differences in beta diversity involve PERMANOVA [44,45] to estimate *p*-values. All *p*-values were adjusted for multiple comparisons with the Benjamini–Hochberg false discovery rate (q = 0.05) [46].

## 3. Results

### 3.1. Sample Characteristics

A total of 527 animals were observed for diarrhea during the group housing phase. Of those, 60% were female (316) and 40% were male (211). Fourteen of the animals whose sample was collected for the analysis had their diarrhea resolve before the third day, so treatment was not initiated and their diarrhea was noted as “transient”. Eight animals were treated with the standard diarrhea protocol (Bismuth Subsalicylate, L’il Critters Brand Gummy Fiber and Gummy Probiotic) and resolved. These animals were categorized as “mild” on comparison measures for microbiome composition. The rest of the animals had diarrhea that persisted or reoccurred after initial treatment (moderate severity), with one animal being euthanized during the observation period for chronic, unresolving diarrhea (severe).

The final set of fecal samples for a 16S analysis included samples from 28 healthy subjects and 33 unhealthy subjects (Table 1). There were slightly more unhealthy samples than healthy samples due to three unhealthy individuals being removed from their group for the observation or treatment of diarrhea before a sample was collected from a healthy animal within their social group. A two-way ANOVA representing the factors of Sex and Health Status revealed no significant effects on age or body weight (see Table 1).

Eleven of the animals with diarrhea had fecal samples evaluated by culture and sensitivity to determine if any bacteria might have caused the diarrhea. Nine of these results were negative for pathogens, and two were positive for *Entamoeba* cysts, normally considered to be non-pathogenic in NHPs [47].

### 3.2. No Differences in Cortisol Levels

Fecal cortisol levels were unchanged across time points (Baseline = Pre; Relocation = Post) in both Healthy and Unhealthy groups of both sexes (Figure 1). Neither the Health Group (Healthy vs. Unhealthy) nor the Time (baseline and relocation) factor showed significant effects. Examinations of the data in Figure 1 suggest that cortisol levels were lower in females with diarrhea after relocation relative to baseline. But, when the factor of Sex was added to the ANOVA, the effect of Time was not significant, nor were any of the interactions.

### 3.3. Animal Relocation Alters the Microbiome Composition

Alpha diversity increased at relocation relative to baseline. This effect was observed with both measures (OTUs and Shannon Index) and in both Healthy and Unhealthy groups (*p* < 0.001; Figure 2). We examined changes in alpha diversity measures from baseline to relocation times but found no effect of Health Status in either measure (Table 2). However, there was a significant effect of Sex as measured by the Shannon Index (Table 2) with males showing a greater change in alpha diversity, *p* < 0.01.

Overarching compositional differences measured by weighted Bray–Curtis and visualized by PCoA were evaluated by Health Status and Time (Figure 3). The spread suggests that Unhealthy animals had a differing microbiome composition than baseline samples of both Health Status groups and a separate one from the Healthy group post-relocation. A Mann–Whitney test indicated a greater beta diversity change between baseline and relocation in Unhealthy subjects (mean: 0.5887) compared to Healthy subjects (mean: 0.5085, W = 125.5, *p* = 0.039).

The filtered results of the One Codex Targeted Loci Database classification were quantified and differences between baseline and relocation times by Health Status group were determined by use of the Mann–Whitney test, with FDR correction (Table 3). Both groups showed an increase in *Actinobacteria* (Healthy group: *p* < 0.03; Unhealthy group: *p* < 0.001) from baseline to relocation. The Healthy group also showed an increase in *Spirochaetota* (*p* = 0.04) after relocation. The Unhealthy group was more enriched with *Heliobacteria* at baseline (*p* = 0.02) than at relocation, and enriched with *Kineothrix* following relocation (*p* < 0.001).

At the species level, the dominant taxa included Collinsella aerofaciens, Prevotella copri, Kineothrix alysoides, Lactobacillus salivarius, Clostridium botulinum, and Lactobacillis reuteri. Two species had significant differences between groups after relocation. Prevotella copri was reduced (*p* < 0.001), and Kineothrix alysoides was enriched (*p* < 0.001) in the Unhealthy group compared to the Healthy group.

## 4. Discussion

This study evaluated the differences in microbiome composition, including alpha and beta diversity, among macaques experiencing a relocation event in an NHP quarantine facility. Assessments were made prior to relocation (e.g., baseline) and after relocation out of the quarantine facilities. Specifically, the later assessments were made when one animal exhibited diarrhea (Unhealthy), at which point fecal specimens were collected from that animal and its matched control (Healthy). We hypothesized that the relocation event would alter the microbiome and that a significant difference in microbiome composition would be noted between samples collected at baseline and samples collected after relocation. Additionally, we explored the hypothesis that the presumed stress of the relocation event would be associated with enhanced fecal cortisol levels. Finally, the sample size was large enough to explore potential sex differences.

The results demonstrated microbiome differences between Healthy and Unhealthy animals. Measures of both alpha and beta diversity differed between baseline and relocation times that were hypothesized to reflect this presumed stress of relocation. However, no differences were found in cortisol levels, an important indicator of animal stress, between the two time points. While there were no sex differences in cortisol levels, we did find some sex differences in microbiome composition. The changes may indicate the role that sex effects play in the digestive health of cynomolgus macaques due to the presumed stress of relocation.

We predicted that the social housing relocation would be an acute stressor, and therefore be associated with differences in cortisol levels post-relocation compared to baseline. However, this was not found in the present study, nor did the onset of diarrhea affect fecal cortisol levels. The cortisol levels at both time levels were higher than what has been reported in zoo primates undergoing relocation stress [48]. Other measurements of stress, such as hair cortisol (a measurement of chronic stress), may provide a more accurate measure than fecal cortisol and could be assessed in future studies.

Social housing transition was associated with alterations in the gut microbiome. Alpha diversity increased in both Health Status groups at the time of relocation compared to baseline. This increase in alpha diversity across both groups after relocation may reflect their exposure to a higher diversity of bacterial taxa in their new environment; we cannot rule out that change in feed and enrichment items between their home country and their current location affected alpha diversity. However, this increase in alpha diversity was associated with sex with males experiencing a greater amount of change between baseline and relocation. These differences suggest the possibility of a sex-dependent effect in health risk in juvenile macaques.

Beta diversity, a measure of sample composition relative to other samples, differed between Unhealthy and Healthy animals. This difference is highlighted in Figure 3, with the distance between Relocation_Diarrhea samples being greater than the Relocation_Control and both baseline groups. A greater change in beta diversity after relocation was observed in Unhealthy subjects. A higher microbial diversity is generally considered beneficial for gut health. For example, microbial diversity is lower in animals with colitis (measured by the Shannon Index) compared to healthy macaques [49,50]. However, while a diverse and abundant gut microbiome is commonly associated with good health, the specific composition and functionality of the microbiome play crucial roles in determining its impact on an individual’s health. Richness and diversity measures can also differ across populations from different geographic origins [22].

Finally, there were some differences in the relative abundance of specific taxa both between Health Status groups and between the two time points. Differences in taxa abundance noted in both groups after relocation included an increase in the phylum *Actinobacteria* [51]. This phylum includes the class *Bifidobacterium*, commonly used as a probiotic to improve health, but it also includes some pathogens [52]. There was also an increase in *Spirochaetota* after relocation but only in the Healthy group.

*Prevotella* was the dominant organism found in the present study. This is consistent with the literature that shows *Prevotella copri* to be the most common amplicon sequence variants found in a colony of free-ranging macaques [53] and in a colony of young rhesus monkeys [54]. *Prevotella* species are normally considered to be beneficial symbionts that assist the host in the digestion of plant material [55] but have also been associated with inflammatory disease and gut dysbiosis [56]. The comparison of the microbiome of wild vs. captive primates demonstrates that the state of captivity results in microbiome changes [57] in which higher relative abundances of *Prevotella* (genus level) and *Bacteriodes* (phylum level) are observed. In the human microbiome, *Prevotella* is a large genus with individual members contributing to different gut functions. This genus has a strong connection to a high-fiber diet [58], shows differences between lean and obese people, is more common in humans who eat non-Westernized diets than in humans who eat a more processed diet [59], and is a feature of dysbiosis shown in autoimmune conditions [60].

Sex differences were noted in measurements of alpha diversity, with males exhibiting a greater loss in alpha diversity after relocation compared to females. Males also had a higher incidence of diarrhea compared to females during the observation period (14:9) for the entire set of 529 animals that were moved out to group housing following quarantine. This is consistent with findings in the human literature indicating a higher number of male patients with irritable bowel disorder compared to females [61]. Sex differences in alpha diversity were also noted in a wild population of immature gorillas, with females showing higher OTU richness [62]. Sex differences in taxa abundance in macaques are also noted in the literature, such as more abundant *Lachnospiracea*, *Bacteriodes*, and *Treponema* in males [49]. In addition, interactions between diet, immune functions, and sex hormones alter the gut microbiome in a sex-dependent manner [63]. Although sex differences in alpha diversity in primate studies could be related to age, steroid hormonal levels, and pubertal transition, the animals in the present study were juveniles with the oldest animals close to the age of puberty. Yet, the role of sex and sex hormones in stress-related alterations in the microbiome warrants further exploration.

A limitation of this study is that we only examined microbiome composition characteristics after the animals’ arrival to the US from Mauritius. The evaluation of fecal samples before shipment to the US would potentially yield more complete information regarding differences between animals who did get diarrhea and ones that remained symptom-free after a relocation event. This timing might also explain why we did not see a difference in cortisol levels, as the animals may not have returned to a baseline state after transport to the US. We assume some similarity of the gut microbiome between individuals because many of them were housed together at their original facility. We examined location as a factor in our analysis in a study after this one and determined that it was not a factor in which animals developed diarrhea. The microbiome samples of animals with diarrhea were compared to animals in their same social housing pen, so sharing the same environment, yet they still differed on the specific measures covered in the analysis. However, controlling for the effect of shared location on microbiome composition was a limitation to address in future studies. It is also possible that using different analyses would lead to results discrepant to those reported here. The use of OTU and alpha diversity indices have limitations. Furthermore, other statistical approaches besides ANOVA have the potential to reveal differing results vis-à-vis the abundance data. However, we have used these analyses in our prior work that was conducted with one of the consultants (K.H.) to this research [64,65]. Finally, the data obtained from the present study provided guidance in the conduct of another study that tested the potential preventative effects of probiotics.

## 5. Conclusions

This study examined the role of social housing changes in microbiome composition after a relocation event and compared animals that did and did not get diarrhea after the relocation. We concluded that gut health was not associated with differences in fecal cortisol levels in this study, but did note that overall, fecal cortisol levels were higher than expected on average across both groups and both time points, indicating a possibility of chronic stress that could be associated with both the initial relocation event (from Mauritius to the US) and then a second relocation event within the facility, from pair housing to group housing with unfamiliar conspecifics. An interesting effect of sex was noted in both the cortisol level data and in diarrhea rates that may indicate a differing health response to stressors based on sex. There was a trend noted of lower cortisol in females with diarrhea compared to males. The overall proportion of males who exhibited diarrhea (*n* = 29, 13.7%) was significantly higher than the proportion of females (*n* = 28, 8.8%). These results suggest differences in the way males and females process stressful events such as social changes. The changes in microbiome composition imply that it would be beneficial to focus efforts on the microbiome changes for future intervention strategies, such as probiotic administration or fecal transfaunation.

## Figures and Tables

**Figure 1 microorganisms-13-00098-f001:**
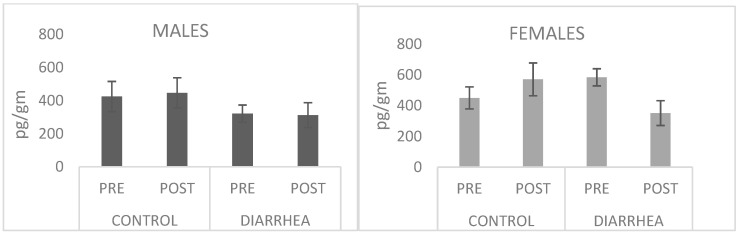
Cortisol levels (pg/gm) by Health Status group (control and diarrhea) and Sex at baseline (PRE) and after relocation (POST).

**Figure 2 microorganisms-13-00098-f002:**
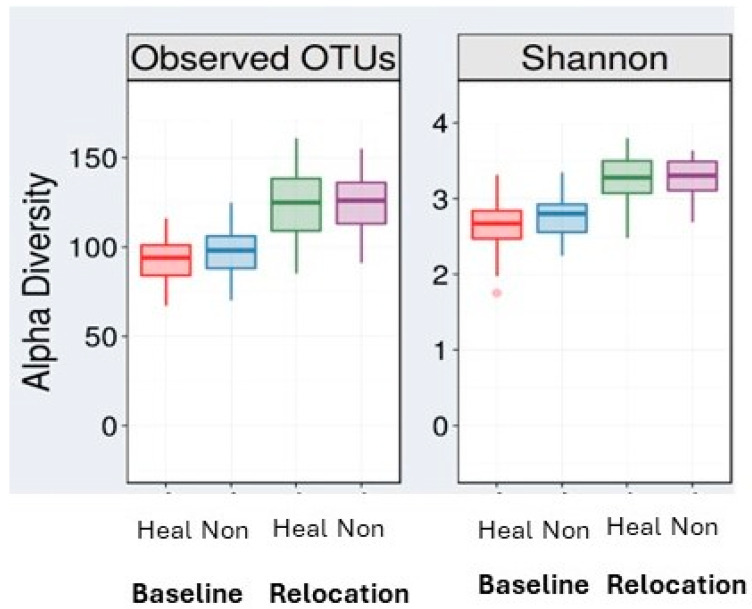
Alpha diversity increases from baseline to relocation in both Healthy (Heal) and Nonhealthy (Non) groups. This effect is seen with both the observed OTUs measured (**Left**) and the Shannon Index (**Right**).

**Figure 3 microorganisms-13-00098-f003:**
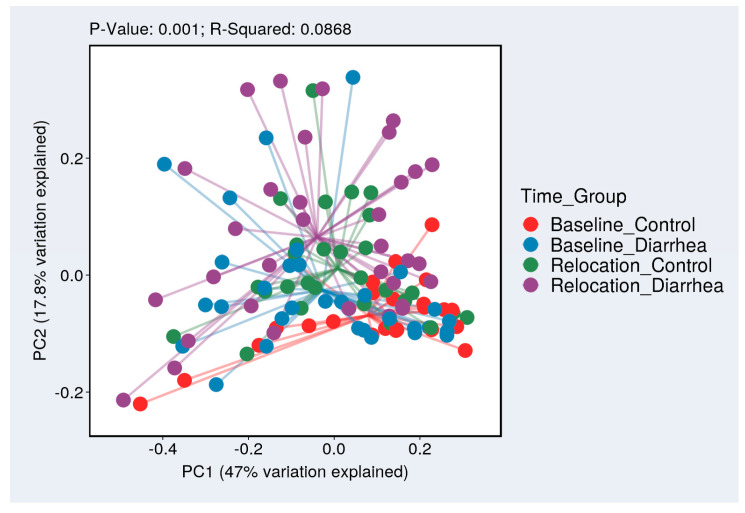
Beta diversity between the four groups of animals illustrated by showing them spread away from a centroid cluster. Red (baseline, Healthy Control samples) and blue (baseline, Unhealthy Diarrhea samples) are clustered together near the bottom right, indicating a similar microbiome composition. Purple (relocation, Unhealthy Diarrhea samples) appears further away from this cluster of red and blue samples, indicating a microbiome composition that differs from baseline samples.

**Table 1 microorganisms-13-00098-t001:** Demographics (n or mean ± S.E.M.) of Animals Expressing Diarrhea (Unhealthy) and Their Matched Controls (Healthy) used for 16S Sequencing.

Measure	Male		Female	
	Healthy	Unhealthy	Healthy	Unhealthy
**Number**	12	13	16	20
**Age (years)**	2.97 ± 0.21	3.00 ± 0.19	3.37 ± 0.17	3.21 ± 0.16
**Body weight (kg)**	3.34 ± 0.21	3.43 ± 0.15	3.29 ± 0.15	3.08 ± 0.17

**Table 2 microorganisms-13-00098-t002:** Change in Alpha Diversity Between Baseline and Relocation by Health Status and Sex.

		OTUs	*p*-Value	Shannon	*p*-Value
**Sex**	**Female**	−23.44	0.10	−0.4778	**0.01**
	**Male**	−34		−0.8084	
**Group**	**Healthy**	−30.41	0.27	−0.6781	0.23
	**Unhealthy**	−24.67		−0.5287	

*Note:* Test statistic used for comparisons—Two-way Repeated Measures ANOVA.

**Table 3 microorganisms-13-00098-t003:** Relative Abundance Comparisons (means) by Group and Time point.

Taxa	Group	Baseline	Relocation	*p*-Value	FDR-Adj. *p*
Bacteroidota	Healthy	0.56	0.50	0.07	0.11
Firmicutes	Healthy	0.38	0.39	0.54	0.54
Actinobacteria	Healthy	0.05	0.07	0.01	**0.03**
Spirochaetota	Healthy	0.01	0.03	0.02	**0.04**
Bacteroidota	Unhealthy	0.48	0.42	0.26	0.26
Firmicutes	Unhealthy	0.42	0.39	0.26	0.26
Actinobacteria	Unhealthy	0.04	0.11	0.00	**0.001**
Spirochaetota	Unhealthy	0.05	0.07	0.13	0.26

*Note:* Test statistics used for comparisons—Mann–Whitney test. Significant findings after FDR correction indicated in bold.

## Data Availability

The data presented in this study are available on request from the first author (K.M.) due to potential propriety rights restrictions.

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
