# Peer review of "Effect of Relocation, Social Housing Changes, and Diarrhea Status on Microbiome Composition of Juvenile Cynomolgus Macaques (Macaca fascicularis)"

_microorganisms, 2025, doi:10.3390/microorganisms13010098_

Round 1
Reviewer 1 Report
Comments and Suggestions for Authors
Line 60-72 are considered to be informations for Materials and methods. Please formulate a specific aim of the study and move the lines 60-72 to the subchapter Materials and Methods. In this form of the manuscript the aim of the study is missing.
Add more details about the animals' diet. Did they receive the same diet from the beginning? During and after the quarantine period? Were there any signs of diarrhea related to the diet? What is the chemical composition of the diet? The nutrient content can influence intestinal transit and the composition of the microbiota.
The results could be explained in more details.
Line 406-407 - mention the sex/gender that presented differences.
Add some new references from 2023-2024
The ideea of the study is very interesting but the final results does not reveale new findings.
Author Response
- Comment: Line 60-72 are considered to be informations for Materials and methods. Please formulate a specific aim of the study and move the lines 60-72 to the subchapter Materials and Methods. In this form of the manuscript the aim of the study is missing. –Response: done
- Comment: Add more details about the animals' diet. Did they receive the same diet from the beginning? During and after the quarantine period? Were there any signs of diarrhea related to the diet? What is the chemical composition of the diet? The nutrient content can influence intestinal transit and the composition of the microbiota. Response: Added more detail about diet lines 295-298.
- Comment: The results could be explained in more details. Response: More details were added.
- Comment: Line 406-407 - mention the sex/gender that presented differences. Response: Added information.
- Comment: Add some new references from 2023-2024. Response: The following citations were added:
Maaskant, A., Voermans, B., Levin, E., de Goffau, M. C., Plomp, N., Schuren, F., ... & Montijn, R. (2024). Microbiome signature suggestive of lactose-intolerance in rhesus macaques (Macaca mulatta) with intermittent chronic diarrhea. Animal Microbiome, 6(1), 53.
-to support incidence of diarrhea/background of problem (line 30)
Populin, L. C., Rajala, A. Z., Matkowskyj, K. A., Saha, S., Zeng, W., Christian, B., ... & Furness, J. B. (2024). Characterization of idiopathic chronic diarrhea and associated intestinal inflammation and preliminary observations of effects of vagal nerve stimulation in a non‐human primate. Neurogastroenterology & Motility, 36(9), e14876.
To support many methods of treatment being investigated (line 31)
Bacon, R. L., Hodo, C. L., Wu, J., Welch, S., Nickodem, C., Vinasco, J., ... & Lawhon, S. D. (2024). Diversity of Campylobacter spp. circulating in a rhesus macaque (Macaca mulatta) breeding colony using culture and molecular methods. mSphere, 9(11), e00560-24.
Cause of mortality, line 33
Bacon, R. L., Taylor, L., Gray, S. B., & Hodo, C. L. (2024). Analysis of cell populations in the normal rhesus macaque (Macaca mulatta) lower intestinal tract and diagnostic thresholds for chronic enterocolitis. Veterinary pathology, 61(2), 303-315.
-idiopathic in origin, line 35
Zhang, Z., Dong, X., Liu, Z., & Liu, N. (2024). Social status predicts physiological and behavioral responses to chronic stress in rhesus monkeys. iScience, 27(6).
Social changes and stress in monkeys line 40
Sarkar, A., McInroy, C. J., Harty, S., Raulo, A., Ibata, N. G., Valles-Colomer, M., ... & Moeller, A. H. (2024). Microbial transmission in the social microbiome and host health and disease. Cell, 187(1), 17-43.
-social, microbiome and immunity, line 43
Rusch, J. A., Layden, B. T., & Dugas, L. R. (2023). Signalling cognition: the gut microbiota and hypothalamic-pituitary-adrenal axis. Frontiers in endocrinology, 14, 1130689.
HPA effects, cortisol release, line 47
Yang, S., Fan, Z., Li, J., Wang, X., Lan, Y., Yue, B., ... & Li, J. (2023). Assembly of novel microbial genomes from gut metagenomes of rhesus macaque (Macaca mulatta). Gut Microbes, 15(1), 2188848.
Reviewer 2 Report
Comments and Suggestions for Authors
Manuscript microorganisms-3344176, entitled “Effect of Relocation, Social Housing Changes and Diarrhea Status on the Microbiome Composition of Juvenile Cynomolgus Macaques (Macaca Fascicularis)”
Recommendation: The above paper is not suitable for publication in its present form.
This article provides information on the effect of relocation, social housing changes and diarrhea status on the microbiome composition of juvenile Cynomolgus Macaques (Macaca Fascicularis). It is in general appropriately organized, carried out and written, however there are some points that should be corrected or clarified.
L9: “…can be associated with…”
L11: “reported” instead of “characterized”
L27-28: “problems” instead of “conditions”
L45: “[11-13]” instead of “[11,12] [13]”
L65-66: What do you mean by “enough samples”?
L82: “evaluated for culture”?
L106: “observed” instead of “seen”
Figure 2: Are these differences significant? P-values?
L152: “…would be associated with…”
L157: “found” instead of “seen”
L178-179: This is not clear in Figure 3
L180: “observed” instead of “seen”
L206: “shown” instead of “noted”
L207: Please delete the second “in”
L236: “Furthermore”
L239: “Finally” instead of “Further”
L247: As shown in Table 1, it is difficult the mean age to be 2.91!
L269-270: Please rephrase
L276: “observed” instead of “seen”
L295: “which animals would break with diarrhea”?
L346: “carried out” instead of “done”
L354-358: This part should be removed to “4.7. Data analysis”
L411: “inform future studies”?
Author Response
- Comment: L9: “…can be associated with…” Response: done
- Comment: L11: “reported” instead of “characterized” Response: done
- Comment: L27-28: “problems” instead of “conditions” Response: done
- Comment: L45: “[11-13]” instead of “[11,12] [13]” Response: done
- Comment: L65-66: What do you mean by “enough samples”? Response: Edited during revision and removed.
- Comment: L82: “evaluated for culture”? Response: added more context
- Comment: L106: “observed” instead of “seen” Response: done
- Comment: Figure 2: Are these differences significant? P-values? Response: P-values were added to the text.
- Comment: L152: “…would be associated with…” Response: done
- Comment: L157: “found” instead of “seen” Response: done
- Comment: L178-179: This is not clear in Figure 3. Response: A more detailed description of the differences were added to the text.
- Comment: L180: “observed” instead of “seen” Response: done
- Comment: L206: “shown” instead of “noted” Response: done
- Comment: L207: Please delete the second “in” Response: done
- Comment: L236: “Furthermore” Response: done
- Comment: L239: “Finally” instead of “Further”Response: done
- Comment: L247: As shown in Table 1, it is difficult the mean age to be 2.91! Response: Clarified that the reported mean in the Subjects and Housing section was at the beginning of the study. Table 1 indicates mean age at the time of diarrhea occurrence, which was 3+ months after arrival (therefore ages had changed).
- Comment: L269-270: Please rephrase Response: added more context around diet (now line 305-310)
- Comment: L276: “observed” instead of “seen” Response: done
- Comment: L295: “which animals would break with diarrhea”? Response: done
- Comment: L346: “carried out” instead of “done” Response: done
- Comment: L354-358: This part should be removed to “4.7. Data analysis” Response: done
- Comment: L411: “inform future studies”? Response: removed this line
- Comment: L9: “…can be associated with…” Response: done
- Comment: L11: “reported” instead of “characterized” Response: done
- Comment: L27-28: “problems” instead of “conditions” Response: done
- Comment: L45: “[11-13]” instead of “[11,12] [13]” Response: done
- Comment: L65-66: What do you mean by “enough samples”? Response: Edited during revision and removed.
- Comment: L82: “evaluated for culture”? Response: added more context
- Comment: L106: “observed” instead of “seen” Response: done
- Comment: Figure 2: Are these differences significant? P-values?Response: P-values were added to the text.
- Comment: L152: “…would be associated with…” Response: done
- Comment: L157: “found” instead of “seen” Response: done
- Comment: L178-179: This is not clear in Figure 3. Response: A more detailed description of the differences were added to the text.
- Comment: L180: “observed” instead of “seen” Response: done
- Comment: L206: “shown” instead of “noted” Response: done
- Comment: L207: Please delete the second “in” Response: done
- Comment: L236: “Furthermore” Response: done
- Comment: L239: “Finally” instead of “Further” Response: done
- Comment: L247: As shown in Table 1, it is difficult the mean age to be 2.91! Response: Clarified that the reported mean in the Subjects and Housing section was at the beginning of the study. Table 1 indicates mean age at the time of diarrhea occurrence, which was 3+ months after arrival (therefore ages had changed).
- Comment: L269-270: Please rephrase Response: added more context around diet (now line 305-310)
- Comment: L276: “observed” instead of “seen” Response: done
- Comment: L295: “which animals would break with diarrhea”? Response: done
- Comment: L346: “carried out” instead of “done” Response: done
- Comment: L354-358: This part should be removed to “4.7. Response: Data analysis” done
- Comment: L411: “inform future studies”? Response: removed this line

Round 2
Reviewer 1 Report
Comments and Suggestions for Authors
The authors responded to all comments and respected all the recommendation